# Chemical Compositions of Chinese Glazed Tiles from an Imperial Mausoleum of the Liao Dynasty

**Lan Zhao** [1,2,*]**, Xiongfei Wan** [3,*]**, Baoqiang Kang** [1,2] **and He Li** [1,2]

1   Department of Conservation Science, The Palace Museum, Beijing 100009, China;
    kangbaoqiang@dpm.org.cn (B.K.); muzili276@hotmail.com (H.L.)
2   Key Scientific Research Base of Ancient Ceramics State Administration of Cultural Heritage, The Palace
    Museum, Beijing 100009, China
3   School of History and Culture, Hubei University, Hubei 430062, China
*   Correspondence: bluesnailan@126.com (L.Z.); 13709822323@163.com (X.W.)

**Abstract:** Glazed tiles are characteristic architectural ceramics traditionally used in ancient Chinese royal buildings. Studies on their chemical compositions have provided valuable information regarding their compositional classifications and the provenances of their raw materials. Existing studies have mainly focused on the Yuan dynasty (1271–1368 AD) or later. Research on earlier ages is limited because of a lack of samples. In this study, we used an energy-dispersive X-ray fluorescence spectrometer to analyze the chemical compositions of 18 glazed tiles unearthed from an imperial mausoleum (the Xinli site) from the Liao dynasty (969–982 AD). The glazes of the tiles had a $SiO_2$–$Al_2O_3$–PbO ternary oxidic system and the bodies of the tiles had a $SiO_2$–$Al_2O_3$ binary oxidic system. Certain compositional differences were observed among the samples with different types of decorations. Compared with samples from the Yuan dynasty and later periods, the Xinli samples had higher $SiO_2$ and $Al_2O_3$ contents and lower PbO and CuO contents in the tile glazes. The tile bodies of the Xinli samples had compositions similar to those of tile bodies from the Qing dynasty (1616–1912 AD). We speculated that the Xinli samples with different decorations came from different kiln sites.

**Keywords:** ceramic composition; provenance study; ancient China

## 1. Introduction

Glazed tiles are characteristic architectural ceramics traditionally used in ancient Chinese royal buildings. Glazed tiles have a long history and can be traced back to as early as the Northern Wei dynasty (386–534 AD) [1]. The glaze formula and the firing steps of glazed tiles were recorded during the Northern Song dynasty (960–1127 AD) in the book of *Ying Zao Fa Shi* (营造法式) [2]. From the Yuan dynasty (1271–1368 AD) to the Qing dynasty (1616–1912 AD), glazed tiles of various colors and quality were widely used in royal buildings and became a distinctive feature [3].

Efforts have been made to reveal the chemical compositions of ancient glazed tiles. This has helped researchers to determine classifications from different times [4–9], as well as the provenances of the raw materials [10,11] and the key parameters of the manufacturing techniques of the glazed tiles [12–15]. Studies have mainly focused on the Yuan dynasty (1271–1368 AD) and subsequent eras because of the abundance of unearthed samples. Research is limited for the era prior to the Yuan dynasty because of the limited availability of tiles. One study on glazed tiles originating prior to the Yuan dynasty was conducted by Shi Ruoyu [5], who explored the raw materials of glazed tiles from the Xixia dynasty (1038–1227 AD) in the northwest of China.

Approximately one century before the Xixia dynasty, the Liao dynasty existed in the northeast of China (907–1125 AD). An imperial mausoleum (the Xinli site, located in Beizhen County, Liaoning Province) was excavated in 2015 [16]. Archaeological data have

revealed that the courtyard building at the Xinli site is the earliest known example of a building with a roof fully covered with green glazed tiles [16]. A number of high-quality green glazed tiles with intact glazes have been excavated, providing abundant samples for research on the raw materials and manufacturing techniques of early Chinese glazed tiles.

We selected glazed tiles (18 pieces) from the Xinli site that were well preserved and decorated with patterns. We used an energy-dispersive X-ray fluorescence spectrometer to analyze the chemical compositions of the bodies and glazes of these tiles to understand the compositional characteristics of the green glazed tiles created during the Liao dynasty and to compare them with those of later ages. We also investigated the provenances of the raw materials used in these glazed tiles via an analysis of the trace elements.

## 2. Materials and Methods

### 2.1. Sample Information

Several glazed tiles were unearthed from the Xinli site [16]. One was decorated with lotus flowers and the rest had an animal-face motif. These glazed tiles were used on the eaves of the buildings (referred to as "goutou" (勾头)) and were often decorated with certain patterns.

Based on the archaeological typology data [17], we selected 18 pieces of well-preserved glazed tiles excavated from the Xinli site (referred to as the Xinli samples). These tiles had an animal-face motif (17 pieces; referred to as the animal-face samples) or a lotus-flower motif (1 piece; referred to as the lotus sample). According to the characteristics of the decorations [17,18], these tiles were grouped into four types (Figure 1 and Table 1).

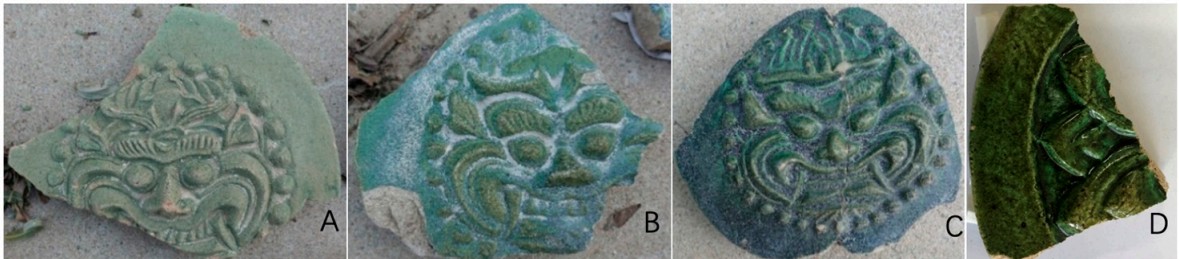

**Figure 1.** Photographs of the samples representing different types of glazed-tile decorations in this study: (**A**) example of type-A animal-face decoration; (**B**) example of type-B animal-face decoration; (**C**) example of type-C animal-face decoration; (**D**) sample with lotus-flower decoration.

**Table 1.** Decoration information on the glazed tiles analyzed in this study.

| Pattern of Decoration | Type | Eyebrow | Forehead | Teeth | Bead Cycle | Ear | Sample Number |
|---|---|---|---|---|---|---|---|
| Animal-face decoration | A | A single shape | A pair of horns in an inverted "eight" shape | 9 incisors | 25 | Flat and wide | 6 |
| | B | Separated eyebrows | A pair of horns with a T-shaped pattern | 7 incisors | 28 | Narrow, long, and slightly bulging | 6 |
| | C | Simplified eyebrows | Straight horns | No clear teeth | 31 | Narrow, long, and bulging | 5 |
| Lotus-flowerdecoration | D | | Double-petal lotus pattern | | | | 1 |

### 2.2. Analytical Method

The glaze and body compositions of these glazed-tile samples were analyzed using an energy-dispersive X-ray fluorescence (ED-XRF) spectrometer (EAGLE III XXL, EDAX Company, Mahwah, NJ, USA), which is a nondestructive analytical method [11,15]. For each sample, the glaze was cleaned and the body was cleaned and polished. The sample was then placed directly into the instrument chamber. The unit was equipped with a rhodium X-ray tube and a Si (Li) detector with an energy resolution of 145 eV at 5.9 keV for Mn–Ka. The equipment was calibrated using a copper sheet. For the ED-XRF measurements, the incident X-ray tube voltage was 25 kV, the current was 600 μA, and the micro-beam size was 0.3 mm. For the major and minor elements, quantitative measurements were achieved via correction and calibration with a set of 13 standard samples.

## 3. Results and Discussion

### 3.1. Compositional Characteristics of the Xinli Samples

The ED-XRF results revealed that the glazes of the Xinli samples had a typical $PbO$–$SiO_2$–$Al_2O_3$ glaze formula. The major elements were PbO (38.4–51.55 wt.%), $SiO_2$ (38.8–45.65 wt.%), and $Al_2O_3$ (4.71–13.52 wt.%) (Table 2). The glazes also contained 0.48–1.11 wt.% CuO as a colorant element, as well as $Na_2O$ (1.3 wt.%, on average), MgO (0.71 wt.%, on average), $K_2O$ (0.67 wt.%, on average), and CaO (0.6 wt.%, on average) as fluxing agents.

**Table 2.** Chemical compositions of the glazes of the Xinli tile samples (wt.%).

| Sample No. | Pattern Type | PbO | $SiO_2$ | $Al_2O_3$ | CuO | $Na_2O$ | MgO | $K_2O$ | CaO | $Fe_2O_3$ |
|---|---|---|---|---|---|---|---|---|---|---|
| 1 | A | 49.53 | 39.16 | 6.44 | 1.11 | 1.38 | 0.55 | 0.51 | 0.55 | 0.41 |
| 2 | A | 43.64 | 40.08 | 11.42 | 0.90 | 1.34 | 0.90 | 0.49 | 0.46 | 0.46 |
| 3 | A | 49.12 | 38.80 | 7.16 | 1.00 | 1.42 | 0.81 | 0.47 | 0.61 | 0.37 |
| 4 | A | 43.59 | 44.26 | 7.99 | 0.80 | 1.23 | 0.54 | 0.52 | 0.35 | 0.36 |
| 5 | A | 45.16 | 39.91 | 9.76 | 0.79 | 1.44 | 0.86 | 0.69 | 0.61 | 0.41 |
| 6 | A | 38.40 | 45.65 | 11.69 | 0.48 | 0.98 | 0.50 | 0.83 | 0.52 | 0.49 |
| 7 | B | 46.16 | 39.76 | 8.70 | 1.05 | 1.44 | 0.80 | 0.58 | 0.66 | 0.42 |
| 8 | B | 43.98 | 39.80 | 10.36 | 1.07 | 1.65 | 0.96 | 0.73 | 0.65 | 0.48 |
| 9 | B | 50.87 | 39.48 | 5.39 | 0.86 | 1.20 | 0.67 | 0.41 | 0.56 | 0.25 |
| 10 | B | 40.35 | 41.21 | 12.79 | 0.75 | 1.44 | 1.00 | 0.72 | 0.91 | 0.53 |
| 11 | B | 40.96 | 43.58 | 9.12 | 0.75 | 1.18 | 0.76 | 1.70 | 0.82 | 0.58 |
| 12 | B | 39.86 | 41.58 | 13.52 | 0.54 | 0.93 | 1.14 | 0.87 | 0.61 | 0.48 |
| 13 | C | 45.95 | 41.32 | 8.09 | 1.08 | 1.26 | 0.37 | 0.60 | 0.57 | 0.44 |
| 14 | C | 50.31 | 39.14 | 5.63 | 0.88 | 1.32 | 0.80 | 0.61 | 0.46 | 0.35 |
| 15 | C | 45.19 | 43.84 | 6.55 | 1.00 | 1.18 | 0.49 | 0.62 | 0.37 | 0.37 |
| 16 | C | 51.55 | 38.95 | 4.71 | 0.95 | 1.36 | 0.81 | 0.43 | 0.34 | 0.42 |
| 17 | C | 50.66 | 40.15 | 5.23 | 0.81 | 1.19 | 0.30 | 0.34 | 0.71 | 0.38 |
| 18 | D | 39.23 | 45.11 | 10.42 | 0.84 | 1.40 | 0.52 | 0.98 | 0.95 | 0.32 |
| Mean | | 45.25 | 41.21 | 8.61 | 0.87 | 1.30 | 0.71 | 0.67 | 0.60 | 0.42 |
| Standard deviation (SD) | | 4.35 | 2.27 | 2.71 | 0.18 | 0.17 | 0.23 | 0.31 | 0.17 | 0.08 |

The major elements of the tile bodies of the Xinli samples were $SiO_2$ (62.12–65.78 wt.%) and $Al_2O_3$ (24.71–28.9 wt.%) (Table 3), indicating that the body samples belonged to the category of high-alumina clay bodies [9]. CaO (1.78 wt.%, on average), $K_2O$ (1.15 wt.%, on average), $Na_2O$ (0.78 wt.%, on average), and MgO (0.69 wt.%, on average) were observed to be fluxing agents. $Fe_2O_3$ (2.17 wt.%, on average) was also observed in the tile bodies.

**Table 3.** Major- and trace-chemical compositions of the bodies of the Xinli glazed-tile samples (wt.%).

| Sample No. | Pattern Type | $SiO_2$ | $Al_2O_3$ | $Na_2O$ | $MgO$ | $K_2O$ | $CaO$ | $TiO_2$ | $Fe_2O$ | $Rb_2O$ | $SrO$ | $Y_2O_3$ | $ZrO_2$ |
|---|---|---|---|---|---|---|---|---|---|---|---|---|---|
| 1 | A | 64.61 | 25.06 | 1.46 | 1.17 | 1.11 | 2.06 | 1.31 | 2.19 | $5.50 \times 10^{-3}$ | $2.93 \times 10^{-2}$ | $2.10 \times 10^{-3}$ | $2.40 \times 10^{-2}$ |
| 2 | A | 65.72 | 25.75 | 0.33 | 0.97 | 0.76 | 1.63 | 1.07 | 2.73 | $5.90 \times 10^{-3}$ | $3.76 \times 10^{-2}$ | $5.00 \times 10^{-3}$ | $3.12 \times 10^{-2}$ |
| 3 | A | 63.32 | 25.91 | 1.52 | 1.06 | 1.14 | 2.58 | 1.10 | 2.33 | $4.10 \times 10^{-3}$ | $3.03 \times 10^{-2}$ | $6.10 \times 10^{-3}$ | $2.89 \times 10^{-2}$ |
| 4 | A | 65.78 | 25.43 | 1.28 | 0.57 | 0.95 | 1.76 | 1.17 | 2.05 | $6.80 \times 10^{-3}$ | $3.48 \times 10^{-2}$ | $4.50 \times 10^{-3}$ | $3.23 \times 10^{-2}$ |
| 5 | A | 63.75 | 27.70 | 0.56 | 0.58 | 1.12 | 1.88 | 1.39 | 2.01 | $4.70 \times 10^{-3}$ | $2.93 \times 10^{-2}$ | $1.30 \times 10^{-3}$ | $2.75 \times 10^{-2}$ |
| 6 | A | 63.82 | 25.99 | 0.99 | 0.89 | 1.31 | 2.03 | 1.28 | 2.62 | $5.60 \times 10^{-3}$ | $2.79 \times 10^{-2}$ | $2.30 \times 10^{-3}$ | $2.53 \times 10^{-2}$ |
| 7 | B | 63.94 | 28.47 | 0.37 | 0.57 | 1.19 | 1.40 | 1.31 | 1.73 | $5.60 \times 10^{-3}$ | $2.44 \times 10^{-2}$ | $2.30 \times 10^{-3}$ | $2.85 \times 10^{-2}$ |
| 8 | B | 65.53 | 24.71 | 0.86 | 0.75 | 1.19 | 2.31 | 1.28 | 2.35 | $2.60 \times 10^{-3}$ | $2.94 \times 10^{-2}$ | $4.00 \times 10^{-3}$ | $3.54 \times 10^{-2}$ |
| 9 | B | 64.28 | 26.35 | 1.06 | 0.78 | 1.12 | 2.01 | 1.25 | 2.11 | $1.60 \times 10^{-3}$ | $2.95 \times 10^{-2}$ | $3.60 \times 10^{-3}$ | $2.59 \times 10^{-2}$ |
| 10 | B | 63.05 | 27.92 | 0.83 | 0.83 | 1.07 | 1.78 | 1.17 | 2.32 | $3.30 \times 10^{-3}$ | $3.13 \times 10^{-2}$ | $5.40 \times 10^{-3}$ | $2.69 \times 10^{-2}$ |
| 11 | B | 64.64 | 27.15 | 0.63 | 0.39 | 0.70 | 1.94 | 1.40 | 2.13 | $5.80 \times 10^{-3}$ | $3.35 \times 10^{-2}$ | $5.20 \times 10^{-3}$ | $2.92 \times 10^{-2}$ |
| 12 | B | 63.84 | 27.76 | 0.38 | 0.41 | 1.50 | 1.83 | 1.21 | 2.05 | $6.40 \times 10^{-3}$ | $3.02 \times 10^{-2}$ | $2.80 \times 10^{-3}$ | $2.76 \times 10^{-2}$ |
| 13 | C | 65.23 | 26.17 | 0.76 | 0.43 | 1.43 | 1.49 | 1.15 | 2.29 | $7.20 \times 10^{-3}$ | $2.26 \times 10^{-2}$ | 0.00 | $2.44 \times 10^{-2}$ |
| 14 | C | 65.39 | 26.96 | 0.72 | 0.42 | 1.20 | 1.33 | 1.27 | 1.68 | $7.10 \times 10^{-3}$ | $1.99 \times 10^{-2}$ | $1.00 \times 10^{-3}$ | $2.18 \times 10^{-2}$ |
| 15 | C | 65.43 | 26.65 | 0.38 | 0.27 | 1.21 | 1.47 | 1.22 | 2.36 | $7.00 \times 10^{-3}$ | $2.65 \times 10^{-2}$ | $6.30 \times 10^{-3}$ | $2.80 \times 10^{-2}$ |
| 16 | C | 62.12 | 27.25 | 0.75 | 0.81 | 1.66 | 1.84 | 1.50 | 3.06 | $8.90 \times 10^{-3}$ | $2.55 \times 10^{-2}$ | $5.30 \times 10^{-3}$ | $3.10 \times 10^{-2}$ |
| 17 | C | 65.49 | 26.74 | 0.67 | 0.39 | 0.91 | 1.74 | 1.14 | 1.89 | $3.10 \times 10^{-3}$ | $2.99 \times 10^{-2}$ | $3.10 \times 10^{-3}$ | $2.36 \times 10^{-2}$ |
| 18 | D | 64.15 | 28.90 | 0.43 | 1.08 | 1.06 | 0.95 | 1.21 | 1.20 | $5.30 \times 10^{-3}$ | $6.30 \times 10^{-3}$ | $4.00 \times 10^{-3}$ | $2.17 \times 10^{-2}$ |
| Mean | | 64.45 | 26.72 | 0.78 | 0.69 | 1.15 | 1.78 | 1.25 | 2.17 | $5.36 \times 10^{-3}$ | $2.77 \times 10^{-2}$ | $3.57 \times 10^{-3}$ | $2.74 \times 10^{-2}$ |
| Standard deviation (SD) | | 1.04 | 1.16 | 0.37 | 0.28 | 0.24 | 0.38 | 0.11 | 0.42 | $1.86 \times 10^{-3}$ | $6.80 \times 10^{-3}$ | $1.83 \times 10^{-3}$ | $3.66 \times 10^{-3}$ |

We then studied the compositional differences in the different decorative types of glazed tiles (Figure 2) to determine the potential differences in the manufacturing techniques, such as the firing temperature [19,20]. When we compared the $Al_2O_3$ contents in the glazes of the samples with different decorative types, we observed a significant difference (*p*-value = 0.036). The type-C samples had lower $Al_2O_3$ contents (mean (SD) = 6.04 (1.33) wt.%) than the type-A (mean (SD) = 9.08 (2.22) wt.%) and type-B (mean (SD) = 9.98 (2.97) wt.%) samples. Therefore, we further classified the samples into two clusters. One comprised the type-A and -B samples, the glazes of which had low contents of lead and high contents of aluminum; the other comprised the type-C samples, the glazes of which had high lead and low aluminum contents. We observed no significant difference in the contents of PbO, $SiO_2$, or CuO. The lotus sample had a lower PbO content and higher $SiO_2$ content than the animal-face decorative samples (Figure 2).

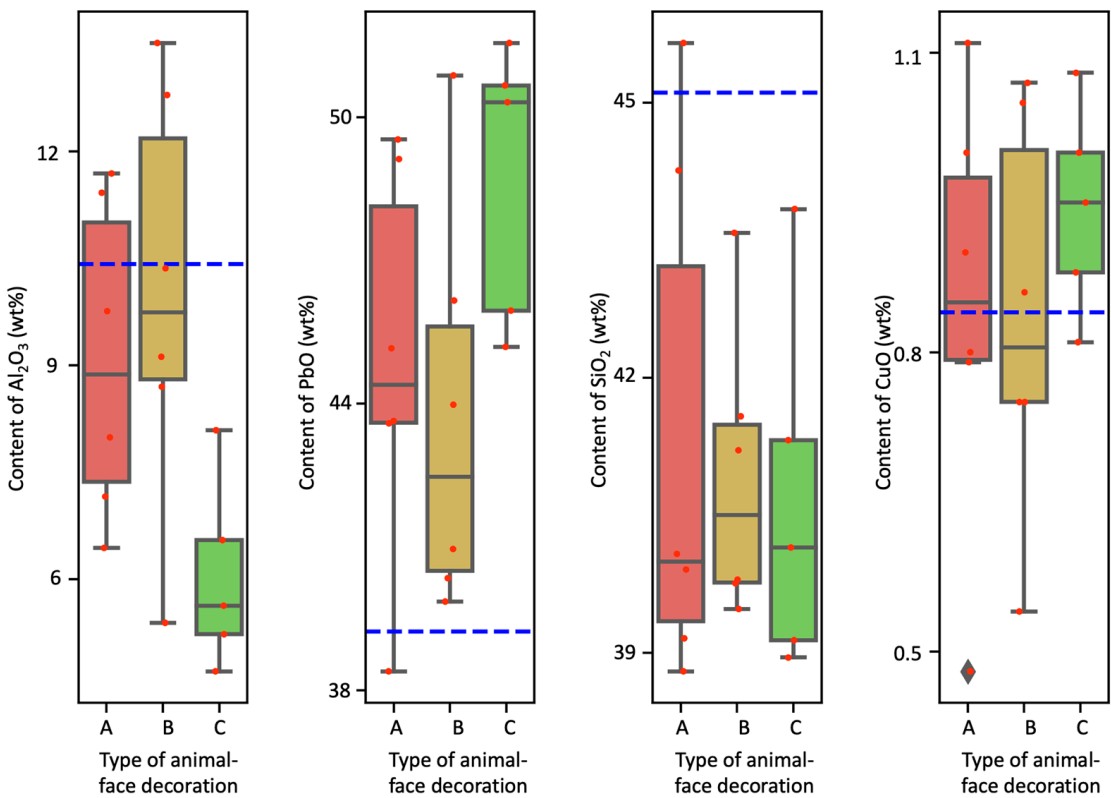

**Figure 2.** Plots of chemical compositions of the tile glazes from the Xinli site with different types of animal-face decorations. Dashed lines represent the corresponding element contents of the lotus-flower-decorated sample from the Xinli site.

There was no significant difference in the major elements (i.e., $Al_2O_3$ and $SiO_2$) in the bodies of the animal-face samples of different types (Figure 3). For the trace elements (i.e., SrO and $Rb_2O$) in the bodies of the three types of samples, we observed a significant difference in the contents of SrO (*p*-value = 0.023). The type-C samples had lower contents of SrO than the other samples. The body of the lotus sample had a higher $Al_2O_3$ content and lower SrO and $ZrO_2$ contents than the bodies of the other samples (Figure 3 and Figure S1).

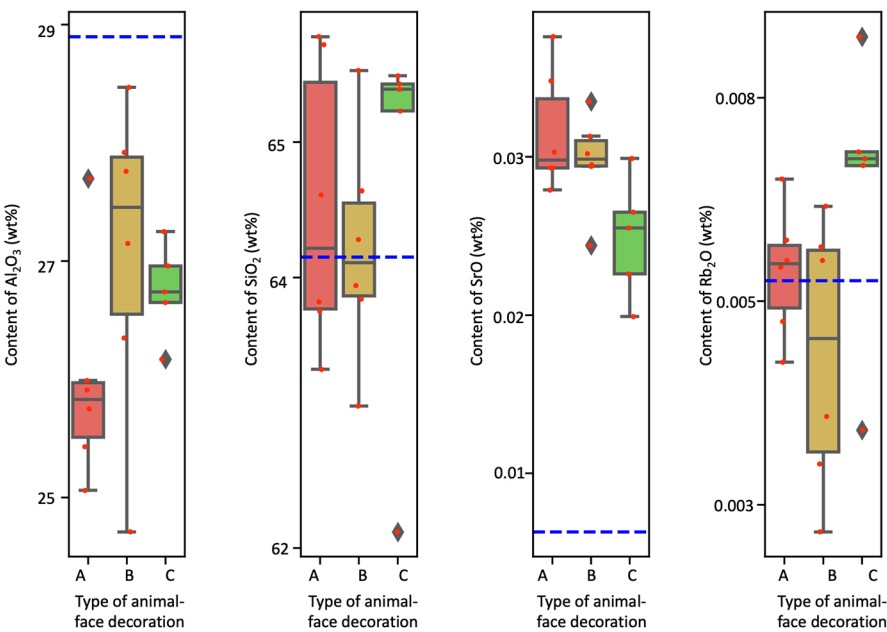

**Figure 3.** Plots of chemical compositions of the body samples from the Xinli site with different types of animal-face decorations. Dashed lines represent the corresponding element contents of the lotus-flower-decorated sample from the Xinli site.

## 3.2. Compositional Difference between the Xinli Samples and Those from Later Periods

To explore the evolution of the chemical compositions of the glazed tiles over time, we compared the samples from the Xinli site (969–982 AD) [3,5–8] with those from later periods (Table 4).

**Table 4.** Descriptions of glazed-tile datasets analyzed.

| Site | Location | Dynasty | Age | Sample Size | Source |
|---|---|---|---|---|---|
| Xinli site | Liaoning Province, China | Liao | 969–982 AD | 18 | - |
| Yuanshangdu site | Neimenggu Province, China | Yuan | 1256–1307 AD | 22 | [5,6] |
| Mingzhongdu site | Anhui Province, China | Ming | 1369–1375 AD | 12 | [7,8] |
| Forbidden City | Beijing, China | Qing | 1616–1912 AD | 2 | [3] |

Regarding the main elements in the glazes, we discovered that the contents of $SiO_2$, PbO, $Al_2O_3$, and CuO in the Xinli samples were significantly different from those in the tiles from later periods (Figure 4). The contents of $SiO_2$ in the Xinli samples were the highest (mean (SD) = 41.21 (2.27) wt.%). The $SiO_2$ contents of tile glazes gradually decreased over time and reached a relatively stable level (27~29 wt.%) in the Ming and Qing dynasties (Figure 4A).

The contents of PbO in the Xinli samples were the lowest (mean (SD) = 45.25 (4.36) wt.%). The PbO contents of tile glazes increased to a stable level of approximately 53 wt.% in the Yuan and Ming dynasties and then reached the highest level (mean (SD) = 60.83 (1.47) wt.%) in the Qing dynasty (Figure 4B). The observed increase in the PbO content of tile glaze over time was consistent with a previous study [4].

The Xinli samples had the highest contents of $Al_2O_3$ (mean (SD) = 8.61 (2.71) wt.%), which was consistent with the study of glazed tiles from the Xixia site [5]. We also observed that the contents of $Al_2O_3$ of tile glazes fluctuated around 3.3~6.4% during later periods.

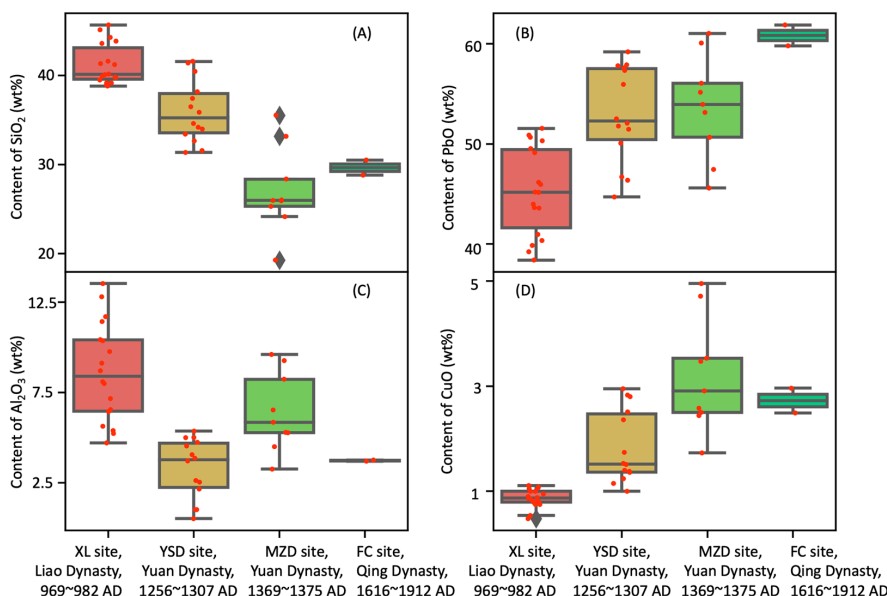

**Figure 4.** Comparison of chemical compositions of the tile glazes from different time periods. XL: Xinli site; YSD: Yuanshangdu site; MZD: Mingzhongdu site; FC: Forbidden City. (**A**) Contents of $SiO_2$; (**B**) contents of PbO; (**C**) contents of $Al_2O_3$; (**D**) contents of CuO.

The contents of the colorant element of CuO increased from 0.87 (0.18) wt.% (mean (SD)) in the Xinli samples to a stable level of ~3 wt.% in the samples from the Ming and Qing dynasties (Figure 4D).

We then compared our results with the major elements in the bodies of glazed tiles from different time periods (Figure 5). The contents of $SiO_2$ in the Xinli samples (mean (SD) = 64.45 (1.04) wt.%) were similar to those of the samples from the Yuan and Ming dynasties (mean (SD) = 62.99 (5.39) wt.% and 62.63 (5.80) wt.%, respectively), but with a smaller variation. The $Al_2O_3$ contents in the Xinli samples (mean (SD) = 26.72 (1.16) wt.%) were higher than those of the samples from the Yuan and Ming dynasties (mean (SD) = 12.24 (2.68) wt.% and 15.96 (1.72) wt.%, respectively), but were close to the samples from the Qing dynasty (mean (SD) = 28.79 (6.89) wt.%). This indicated that these samples were typical of the high-$Al_2O_3$-content tile bodies from northern China and that they were fired at a higher temperature than those with lower $Al_2O_3$ contents [9].

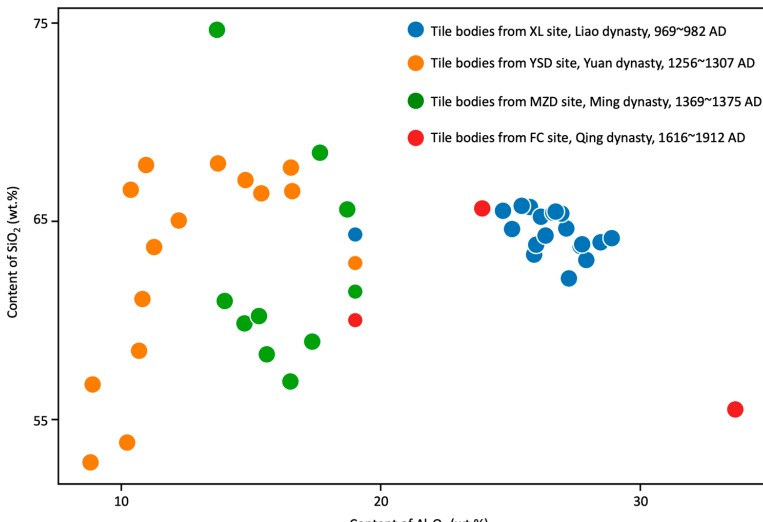

**Figure 5.** Scatterplots of the $SiO_2$ and $Al_2O_3$ contents of tile bodies from different time periods. Each dot represents a tile-body sample. XL: Xinli; YSD: Yuanshangdu; MZD: Mingzhongdu; FC: Forbidden City.

### 3.3. Provenance Study of the Xinli Samples

To investigate the provenances of the glazed tiles from the Xinli site, we compared their trace elements with those from two other ancient kiln sites of the Liao dynasty. One was the ancient kiln site in Gangguantun, Liaoning Province (Table S1); the other was the Gangwa kiln site in Chifeng, Neimenggu Province, which was the capital of the Liao dynasty [21].

The animal-face decorative samples from the Xinli site (Figure 6) had similar contents of $Rb_2O$ and SrO to those of the Gangwa samples. An unfinished tile with a pattern similar to those of the Xinli type-C samples was unearthed at the Gangwa kiln site (now held by the museum of Liaoning Province) [18]. This indicated that the animal-face-decoration samples from the Xinli site may have been produced at the Gangwa kiln site.

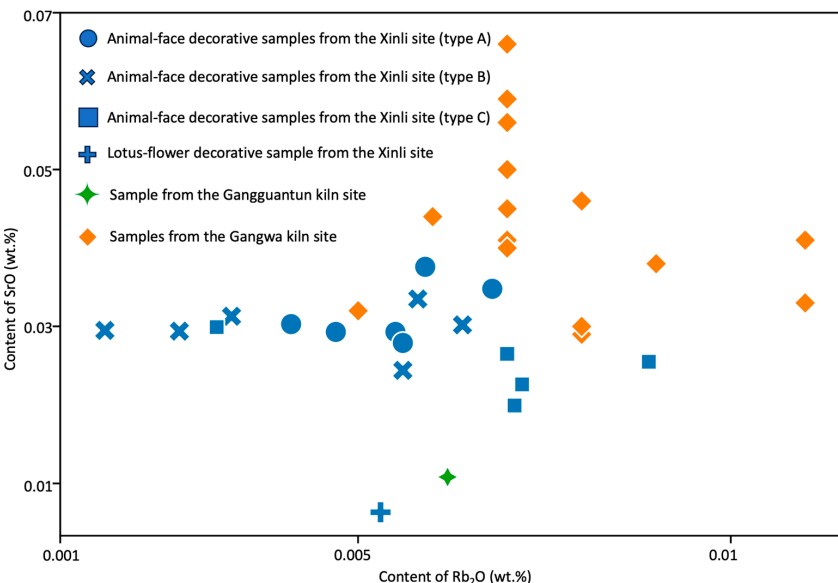

**Figure 6.** Scatterplots of the SrO and $Rb_2O$ contents of the tile bodies from different kiln sites of the Liao dynasty. Each dot represents a tile-body sample.

We observed that the lotus sample from the Xinli site and the Gangguantun sample had similar contents of $Rb_2O$, SrO, $Y_2O_3$, and $ZrO_2$ (Figure 6 and Figure S2). This indicated that the lotus sample may have been produced at the Gangguantun kiln site. Our trace-element-difference findings between the differently decorated Xinli samples echoed the hypothesis that different decorative patterns may represent characteristics of different periods of the Liao dynasty [17].

We also observed different patterns in the trace-element contents for the different types of animal-face-decoration samples from the Xinli site. The type-C samples had higher contents of $Rb_2O$ and lower contents of SrO than the type-A and -B samples. We conjectured that these samples may have been produced at the same kiln site, but by different manufacturers.

### 4. Conclusions

The bodies of the glazed tiles from the Xinli site had a $SiO_2$–$Al_2O_3$–PbO ternary oxidic system in which the total contents of $SiO_2$, $Al_2O_3$, and PbO were greater than 90%. The glazes had a $SiO_2$–$Al_2O_3$ binary oxidic system in which the total contents of $SiO_2$ and $Al_2O_3$ were as high as 90%. Several compositional differences (e.g., the content of $Al_2O_3$ in the glaze formula) were also observed between the samples with different types of decorations.

The chemical compositions of the glazed tiles from the Xinli site (Liao dynasty, 969–982 AD) were somewhat different from those of the glazed tiles from sites of later periods. The contents of $SiO_2$ and $Al_2O_3$ in the tile glazes of the Xinli samples were relatively high, whereas the contents of PbO and CuO were relatively low. We observed that the $Al_2O_3$ and $SiO_2$ contents in the tile bodies from the Xinli site were different from

those in the tile bodies from the Yuanshangdu site (1256–1307 AD) and Mingzhongdu site (1369–1375 AD), but similar to those in the tile bodies from the Forbidden City (1616–1912 AD). A compositional characterization of the Xinli samples would help to infer the firing temperature and other relevant information from their manufacturing process.

Through a comparison of the trace elements in the body tiles from the Xinli site and those from the body tiles of other kiln sites of the Liao dynasty (i.e., the Gangguantun kiln site and the Gangwa kiln site), we observed that the Xinli samples with animal-face decorations had similar contents of $Rb_2O$ and $SrO$ to those from the Gangwa kiln site. We also discovered that the Xinli sample with the lotus decoration had similar contents of $Rb_2O$ and $SrO$ to those of the sample from the Gangguantun kiln site. These findings indicated that the glazed tiles with different decorations might have been produced at different kiln sites. There were differences in the $SrO$ contents of the samples with different types of animal-face decorations; we conjectured that these samples might have been produced at the same kiln site, but by different manufacturers.

The glazed tiles analyzed in this study have been exposed to an open environment for hundreds of years. The chemical compositions of these samples may differ from their original forms due to degradation or the leaching of lead or other cations. Our findings on the compositional patterns of these samples were robust to this bias. We will continue to study changes in the chemical compositions in association with the exposure time and the leaching of sodium, lead, and other cations to obtain more accurate results.

**Supplementary Materials:** The following supporting information can be downloaded at https://www.mdpi.com/article/10.3390/ceramics6040123/s1: Table S1: Major and trace elements of a sample tile body from the Gangguantun kiln site, Liaoning Province (wt.%); Figure S1: Plots of $Y_2O_3$ and $ZrO_2$ contents in body samples from the Xinli site with different types of decorations; Figure S2: Scatterplots of the $Y_2O_3$ and $ZrO_2$ contents of tile bodies from different kiln sites of the Liao dynasty.

**Author Contributions:** Conceptualization, L.Z. and X.W.; methodology, L.Z.; software, L.Z. and H.L.; validation, L.Z.; formal analysis, L.Z.; investigation, L.Z.; resources, X.W.; data curation, L.Z.; writing—original draft preparation, L.Z.; writing—review and editing, L.Z.; visualization, L.Z.; supervision, L.Z.; project administration, B.K. All authors have read and agreed to the published version of the manuscript.

**Funding:** This work was supported by the National Social Science Fund of China (grant numbers 18ZDA226 and LSYZD21019) and by the Key Base of Humanities and Social Sciences of Colleges and Universities in Jiangxi Province, China (grant number JD15112). The funders had no role in the study design, data collection, analysis, decision to publish, or manuscript preparation.

**Institutional Review Board Statement:** Not applicable.

**Informed Consent Statement:** Not applicable.

**Data Availability Statement:** All of the analyzed data are available within the article.

**Acknowledgments:** The authors are grateful to Jian Zhang and Yunjun Zhang at Peking University for the insightful discussions and the careful examination of the manuscript. The authors are grateful to the editor and reviewers for their constructive comments, which have significantly improved the quality and clarity of the manuscript.

**Conflicts of Interest:** The authors declare no conflict of interest.

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
