# Peer review of "Chemical Compositions of Chinese Glazed Tiles from an Imperial Mausoleum of the Liao Dynasty"

_ceramics, doi:10.3390/ceramics6040123_

Round 1
Reviewer 1 Report
The article is very interesting. The chemical composition of these ancient tiles is the focus of the investigation, yet the degradation of the samples surface has not been mentioned in terms of Na (or other cations) leaching.
Abstract
Line 18: Could you rephrase the sentence: "to have SiO2-Al2O3-PbO formula and the body have SiO2-Al2O3 formula"? formula is the wrong term, it would be preferred: ternary and binary oxidic systems, respectively.
Introduction
Lines 51-55: Could you motivate the use of EDS analyses? Was it used in a non destructive way? or was it destructive? Micro-sampling was adopted? Could you mention here the influence of the degradation degree of the tiles on their chemical analysis? Previous use of EDS in ancient glazed tiles should be reported. Leaching of Na and other monovalent cations out of the glaze or ceramic structure are very heavy thus leading to overestimation of SiO2 and Al2O3 in several cases.
2.1. Sample information
Table 1, last column: Is it correct to indicate "Sample Size"? Could it be better replaced with: "Sample number"?
2.2. Analytical Method
Lines 80-81: Please complete the description of the analytical method by indicating sample preparation. The entire piece of tile was inserted in the ED-XRF? How was the surface prepared? Chemical composition of ancient tiles can be affected by degradation, thus typically the samples are polished to obtain a clean and flat surface before EDS analysis. Please check the Ref: Minerals 2020, 10(3), 272; https://doi.org/10.3390/min10030272.
4. Conclusions
Lines 190-191: Leaching of some cations from glaze aluminosilicate network should be mentioned here.
Author Response
Thanks for your careful reviewing and constructive comments, which have significantly improved the quality and clarity of our manuscript. Please see the attachment for our response.

Reviewer 2 Report
In the manuscript, the author have collected several samples of glazed tiles from the ancient sites and performed their compositional studies using XRF technique. Mainly the wt. % of silica/SiO2, alumina/Al2O3, glazing-agent/PbO and colorant/CuO in tiles’ glaze and body are chronologically studied in samples obtained from the same and/or different sites. Authors used provenance of obtained samples to establish a compositional variance within the set of samples. Though the authors presented results nicely, but their analysis/conjecture has some serious flaws, hence I suggest a major revision of manuscript.
L20 “the Xinli samples had higher SiO2 and Al2O3 contents (cements) and lower PbO and CuO (agents) contents in the tile glaze”. Table 2 shows roughly the same ratio 1:1 of silica/alumina in the pottery cement which leads to poor tile strength and lower glaze among the tiles. How does it relate to some different kiln technology as claimed by authors? May be in the later dynasties the potters were simply improving the ratio to increase tile quality.
L21 “The tile body of Xinli samples have similar composition to those from Qing Dynasty”. Authors should comment on how such compositional change could lead to a higher tile strength. Table 3 shows silica/alumina ratio is 2.6 similar to the modern cement ratio; the modern ratio in range of 2.5 to 4 gives maximum mechanical strength of pottery. Moreover, author is suggesting (L152), “the (kiln) technology of making tile body in Xinli was close to that of Qing Dynasty”. Such claims would remain unconfirmed unless the author attempts to measure the mechanical strength of those two sets of samples. May be the potter was simply trying to make a stronger pottery hence changing the silica/alumina ratio, which has nothing to do with the kiln technology.
L195 “These revealed changes in chemical composition over time also help to shed light on the evolution of glazed tile manufacturing techniques”. Authors simply claim that the change in composition leads to change in manufacturing techniques, which is again unsubstantiated unless some details of the kiln methods are presented
I suggest performing additional study of mechanical properties on various sets of samples to find out how mechanical strengths of the tiles were evolved and compare them with the systematic improvement in the glaze of the sample taken from the ancient kilns. This measurement is necessary to justify the conclusions, of change in technology, given in the manuscript.
Author Response

(The authors gave the same response as above.)

Reviewer 3 Report
The article is well written and everything is clear. However the plots (figures 5 and 6) don’t show clear or unique associations. In figure 6, the samples from Gangwa Kiln site identify a define a strange linear direction of data determined by an equal content of Rb in comparison to Sr. The authors should give additional information about these trends, and use others plot to demonstrate their hypotheses. The population of samples is very low, so the authors should combine others analytical investigations to have a real comparison of samples (for example spectrocolorimeter, XRD). For this reason, I invite the authors to improve the article adding more data also taking the results present in literature to improve the statistic.
Author Response

(The authors gave the same response as above.)

Reviewer 4 Report
This is a good study, with appropriate methods and interpretation. My one concern is that it is based on 18 glazed tiles, with just one a lotus flower. How/why did you do just 18?
Check line 58, you say "three" being lotus flowers, I suspect that is incorrect.
In Table 2, and the associated text, you have two decimal places, but for the PbO, SiO2, and Al2O3, you should have just three digits given the precision of the measurements.
Capitalize first letter in words in Table 1.
Be consistent with periods after "wt"
You report trace element data for Y, Zr but only deal with Sr and Rb in the text. Why - not useful?
Check citation numbering sequence in part 3.3

Throughout you need to add an "a" or "the" - see the markup on the attachment. Also some extra spaces.
Author Response
Thanks for your careful review and constructive comments, which have significantly improved the quality and clarity of our manuscript. Please see the attachment for our response.

Round 2
Reviewer 2 Report
All of my previous comments are suitably addressed by the authors. Recommended for the publication.
Reviewer 3 Report
Thanks for your revised paper.